# Mechanism of effector capture and delivery by the type IV secretion system from *Legionella pneumophila*

Amit Meir [1,2,4,6✉], Kevin Macé[1,6], Natalya Lukoyanova [1], David Chetrit[2], Manuela K. Hospenthal [1,5], Adam Redzej [1], Craig Roy [2✉] & Gabriel Waksman [1,3✉]

*Legionella pneumophila* is a bacterial pathogen that utilises a Type IV secretion (T4S) system to inject effector proteins into human macrophages. Essential to the recruitment and delivery of effectors to the T4S machinery is the membrane-embedded T4 coupling complex (T4CC). Here, we purify an intact T4CC from the *Legionella* membrane. It contains the DotL ATPase, the DotM and DotN proteins, the chaperone module IcmSW, and two previously uncharacterised proteins, DotY and DotZ. The atomic resolution structure reveals a DotLMNYZ hetero-pentameric core from which the flexible IcmSW module protrudes. Six of these hetero-pentameric complexes may assemble into a 1.6-MDa hexameric nanomachine, forming an inner membrane channel for effectors to pass through. Analysis of multiple cryo EM maps, further modelling and mutagenesis provide working models for the mechanism for binding and delivery of two essential classes of *Legionella* effectors, depending on IcmSW or DotM, respectively.

[1] Institute of Structural and Molecular Biology, Birkbeck and UCL, Malet Street, London WC1E 7HX, UK. [2] Boyer Center for Molecular Medicine, Department of Microbial Pathogenesis, Yale University, 295 Congress Avenue, New Haven, CT 06536-0812, USA. [3] Institute of Structural and Molecular Biology, University College London, Gower Street, London WC1E 6BT, UK. [4] Present address: Boyer Center for Molecular Medicine, Department of Microbial Pathogenesis, Yale University, 295 Congress Avenue, New Haven, CT 06536-0812, USA. [5] Present address: Institute of Molecular Biology and Biophysics, Department of Biology, ETH Zürich, Otto-Stern-Weg 5, 8093 Zürich, Switzerland. [6] These authors contributed equally: Amit Meir, Kevin Macé. ✉email: a.meir@mail.cryst.bbk.ac.uk; craig.roy@yale.edu; g.waksman@mail.cryst.bbk.ac.uk

Legionella pneumophila is an opportunistic human pathogen that causes a type of severe pneumonia called Legionnaire's disease[1]. It also has natural hosts among protozoa[2]. The bacterium translocates into the cytosol of the host a plethora of effector proteins that hijack cell functions to create a specialised organelle that supports intracellular replication[3]. L. pneumophila effectors are injected into the host using a T4S system[4,5], which is encoded by 27 genes of the dot/icm gene cluster including 3 ATPases namely DotO, DotB, and DotL[6]. In the study presented here, we focus on L. pneumophila DotL, a membrane-embedded AAA + T4S system ATPase and member of the VirD4 family of proteins[7]. In Legionella, DotL is part of a large complex that includes the proteins DotM and DotN (Supplementary Fig. 1a). Together, they form the Type IV coupling complex (T4CC)[8]. The T4CC contains several types of binding sites that recruit different classes of effectors. Indeed, depending on their mode of recruitment, effectors can be grouped into two classes: (a) effectors that are dependent on their binding to a complex of the proteins IcmS and IcmW (IcmSW)[9], and (b) the other that include effectors which are IcmSW-independent. Among the latter, there is a subset of effectors that contain a C-terminal secretion signal sequence rich in Glu residues (referred to as "Glu-rich signal peptide or Glu-rich SP")[10]. Effector-bound IcmSW binds to T4CC through binding to the very C-terminal sequence of DotL (Supplementary Fig. 1a)[11–13]. In contrast, the subset of acidic Glu-rich SP-containing effectors bind to DotM[14]. Although some single component fragments of the T4CC are structurally characterised[12,14], there is no overall view of the intact, fully assembled complex. Here, we present the atomic structure of an intact, fully-assembled Legionella T4CC.

## Results and discussion

**Purification and composition of the T4CC.** We purified the T4CC from Legionella cell membranes after solubilisation with detergents and taking advantage of a Strep-tag inserted at the C-terminus of DotL (Methods and Supplementary Tables 1, 2 and 3). The complex not only contains DotL, DotM and DotN but also 5 additional proteins (Fig. 1a): IcmS, IcmW, LvgA, and two previously-uncharacterised proteins encoded by two annotated open reading frames, lpg0294 and lpg1549. We named these proteins DotY and DotZ, respectively, since they co-purify with the dot/icm Legionella T4CC. The size of this complex is ~300 kDa as assessed by SEC-MALS, consistent with a complex that may contain 1 copy each of the 8 proteins. The presence of DotY and DotZ (the encoding genes of which are located outside the Legionella dot/icm gene cluster) was unexpected. To assess their role, three deletion mutants were made, ΔdotY, ΔdotZ and ΔdotYZ, where the dotY or dotZ or both genes were deleted, respectively. Intracellular growth of these mutants in the protozoa Acanthamoeba castellanii was reduced (47(±11), 36(±26), and 25 (±15)%, respectively, compared to wild-type), consistent with a previous report of a transposon-insertion lpg0294 (dotY) mutant[15]. Complementation of the ΔdotY or ΔdotZ strains with wild-type dotY or dotZ gene, respectively, restored growth to wild-type levels (Supplementary Fig. 2), indicating that reduction in intracellular growth in the mutants is due to deletion of the targeted gene(s). These data are similar to results obtained for the mutants that are deficient in the coupling protein chaperone proteins IcmS, IcmW, and LvgA[16,17]. These mutants display relatively minor defects in effector translocation assays, that result in more pronounced defects in intracellular replication. Since the intracellular replication defects are the result of decreased effector translocation, complementation studies typically use intracellular replication assays to confirm the absence of a secondary

mutation, which confirms that effector translocation defects have been restored.

Next, effectors translocation was monitored using Cya-fusions of 5 different effectors: RalF, a well-characterized effector, Lem21 and LegC8, which are known IcmSW-dependent effectors[14], Lpg1663 and CegC3 which are acidic Glu-rich SP-containing effectors that we have recently shown to be recruited by DotM[14]. We show here that translocation of these effectors into CHO cells was affected significantly by dotY or dotZ deletions (5- to 10-fold reduction compared to wild-type (Fig. 1b)). The Intracellular growth complementation results confirm the decreased levels of effector translocation are due to the genes deletions and not due to secondary mutations.

Finally, we show that DotY and DotZ are not themselves translocated (Fig. 1c). We conclude that DotY and DotZ are integral parts of the T4CC and play significant roles in the translocation of most effectors that we have tested. They are also unique to the Legionella genus.

**Structure of the DotLMNYZ hetero-pentameric complex.** The structure of the T4CC was next determined using cryo-electron microscopy (cryo-EM). Single particle reconstruction yielded a map with an average resolution of 3.7 Å but 3.5 Å in many parts (Supplementary Fig. 1b–f). The quality of the electron density was sufficient (Fig. 1d) to solve the structures of DotL and DotM except for their trans-membrane segments, DotN, DotZ, and the first N-terminal 77 residues of DotY (Figs. 1e and 2, and Supplementary Table 4). These parts constitute the "hetero-pentameric core" of the T4CC. This core structure has the shape of a right angle triangle with a longer and short side of 14.2 and 9.1 nm, respectively (Fig. 1e). The transmembrane segments of DotL and DotM locate at the end of the short side (Fig. 1e). The structure of the T4CC hetero-pentameric core reveals a large interaction network involving 18,146 Å² of buried solvent accessible surface area, bringing all 5 proteins together through 8 interfaces (Figs. 2 and 3, and Supplementary Figs. 3 and 4).

DotL plays a central role in the assembly of the T4CC, forming the 1st and 2d largest interfaces, with DotM (3089 Å² in each protein) and DotN (1756 Å²), respectively. DotL has a long C-terminal tail which starts at residue 572 in the region of the structure proximal to the membrane just after the β11 strand and runs down the entire T4CC structure (Figs. 1e, 2 and 3a–d, and Supplementary Figs. 3a and 4b–d). It encompasses a long segment termed "β11α14" (Fig. 3b, c), which is so-called because it lies between β11 and α14 (thereafter, all regions between secondary structures will be referred to in a similar way). This region of the tail is an integral part of the DotL structure and makes extensive interactions with DotM, notably running within a groove formed by two sub-domains of DotM (Fig. 3c). It is followed by helix α14, a loop between α14 and α15, and finally α15 and α16: this region forms the interface with DotN[12] (Fig. 3d and Supplementary Figs. 3c and 4c). Other regions of DotL involved in DotM binding include residues in the region of the β1 and β2 strands, residues in the α2β4, α3β5 and β5α4 regions, as well as in the region at and around α10 (α9α10 and α10β6) and in α11β7. On the DotM side, residues in contact with DotL are situated in the α1 to α3 region at the N-terminus of the protein, and in the C-terminal half of the protein from η1 to α14 (Fig. 3b, Supplementary Figs. 3a, b and 4b). Likewise, the interactions stabilising the recruitment of DotN to the T4CC core extend beyond the DotL C-terminal tail. DotN is indeed further involved in contact with DotM, together forming with this protein the 5th largest interface between T4CC core proteins (659 Å² in each protein). Residues in α4 and α12α13 regions of

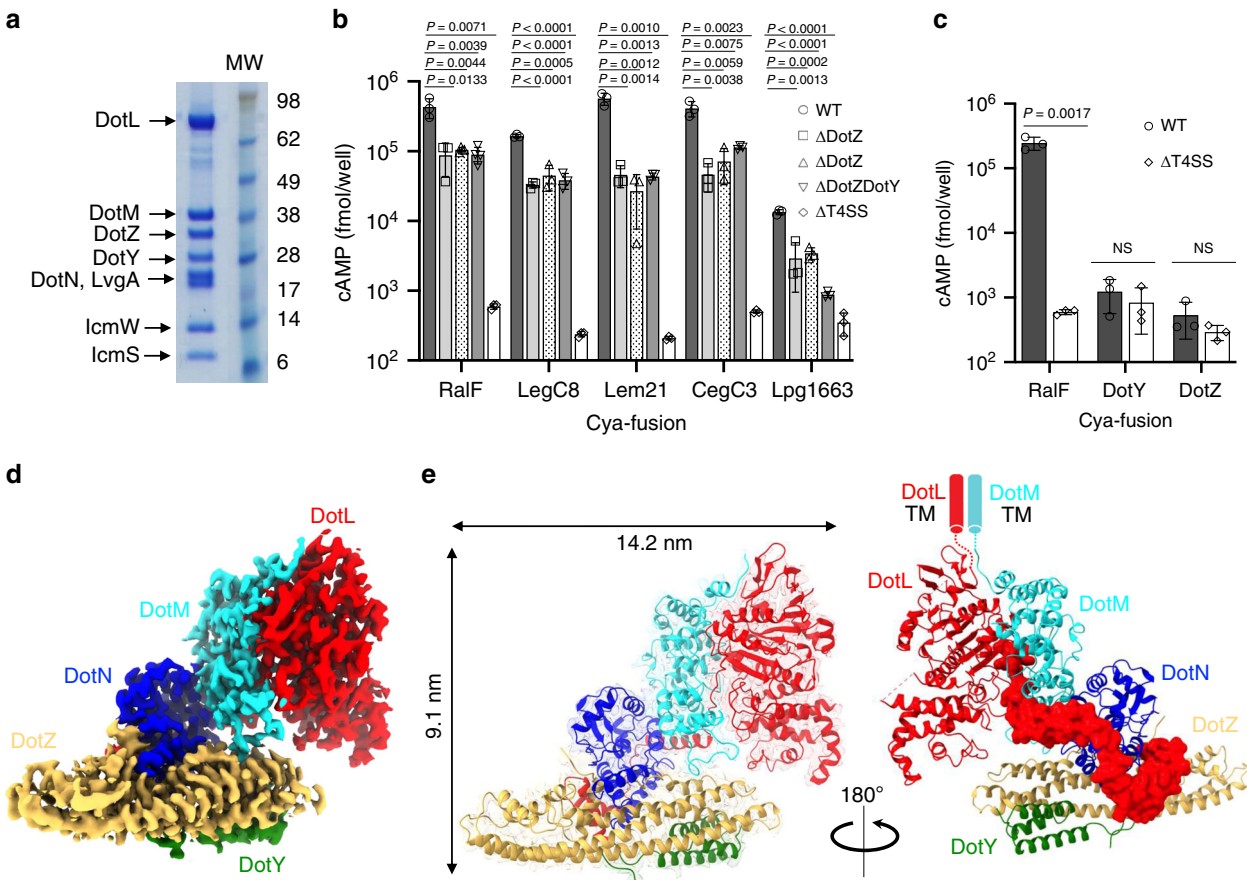

**Fig. 1 Biochemical, biological and structural characterisation of the *Legionella* T4CC. a** SDS-PAGE analysis of the purified complex. Lane 1: the T4CC, Lane MW: molecular weight markers. Protein bands were identified by mass spectrometry and are labelled accordingly. Molecular weights in lane MW are also provided. SDS-PAGE analysis of the complex was routinely carried out after each preparation (at least seven times over the course of the project) and yielded the same result. A control pull-down using an untagged version of DotL did not result in any complex being purified (result not shown). **b** Role of DotY and DotZ in effector translocation. Levels of translocation by the deletion mutants ΔdotY, ΔdotZ and ΔdotYZ were compared to the Lp01 wild-type strain (WT) and a strain defective in the T4S system (ΔT4SS)[45]. Bar shape-coding for each mutant and wild-type strains is indicated on the right. For the ΔdotY, ΔdotZ, ΔdotYZ mutants, differences in translocation levels were found to be significant with a *P* value of <0.005 comparing to WT. Only for RalF, *P* values were <0.05. **c** Translocation of DotY and DotZ. Translocation of Cya-DotY and Cya-DotZ was compared to Cya-RalF, and also assayed in the ΔT4SS strain. Bar color-coding is shown at the top. For both Cya-DotY and Cya-DotZ, there were no significant differences in translocation levels between the WT and ΔT4SS strains. **d** Electron density of the T4CC hetero-pentameric core. The map was contoured at 7 σ level. Color-coding is per protein, red, cyan, blue, orange yellow and green for DotL, DotM, DotN, DotZ and DotY, respectively. **e** Two views of the structure of the T4CC hetero-pentameric core. The two views are related by 180° rotation. At left, DotL, DotM, DotN, DotZ and DotY are shown in ribbon, color-coded red, cyan, blue, orange yellow, and green, respectively. Electron density is as in panel d, but semi-transparent. Complex dimensions are indicated. At right, the proteins are represented in ribbon except for the C-terminal tail of DotL, which is shown in surface representation. Color-coding is as shown in the panel at left. The locations of the disordered transmembrane (TM) regions of DotM and DotL are indicated by a cyan and red cylinders, respectively, thereby providing the location of the inner membrane (IM). For (**b**) and (**c**), data are representatives of three independent experiments (*n* = 3), each strain with biological triplicates. For each independent experiment, effector translocation values in mutants were normalized against their ratio to WT. Graphs report mean intracellular cAMP levels ± standard deviation for each strain. Indicated *P* values are mutant strains in comparison to wild-type, calculated by two-tailed Student's *t* test. NS not significant (*P* > 0.05). For (**a-c**), Source data are provided as a Source Data file.

DotM interact with residues in α1, α1α2, α3β3, and α6 of DotN (Fig. 3e and Supplementary Figs. 3b, c and 4e).

DotZ lies along the long side of the T4CC's right triangle shape. This 294 amino acid protein has an elongated structure, the core of which is made of three long α-helices (α3, α4, and α7) (Figs. 2 and 3f). Four smaller helices, α1, α2, α5, α6 wrap around the distal end of the α3–7 bundle, α1 departing at a 90° angle to form together with α3 a groove (Fig. 3f). It makes significant contact with DotN (1517 Å², the 3rd largest interface) and DotY (1396 Å², the 4th largest), but engages weakly with DotM (355 Å²) and DotL (177 Å²) (Supplementary Figs. 3a–e and 4d, f–h). DotN interactions with DotZ include residues of the very C-terminal helix (α8) and residues in α4 and the α4α5 loop of DotN

which inserts into the DotZ groove (Fig. 3g). DotM interacts with DotZ through an extended loop, β1β2. β1β2 of DotM, α7 and α8 of DotZ, and α3β3 of DotN form a cavity (Figs. 2 and 3h and Supplementary Fig. 4h).

For DotY, clear density is only observed for three N-terminal helices (α1–3) and a substantial loop following α3 (Figs. 2 and 3i). These regions of DotY are ordered because they make extensive interactions with DotZ. This interface brings together residues in α1 and α2 of DotY with residues in α3 and α4 of DotZ and a cluster of residues in the DotY α3 loop with the DotZ groove (Fig. 3i and Supplementary Figs. 3d,e and 4g). Proximity of DotN and DotY within the DotZ groove leads to interactions between these two proteins (Supplementary Fig. 4i).

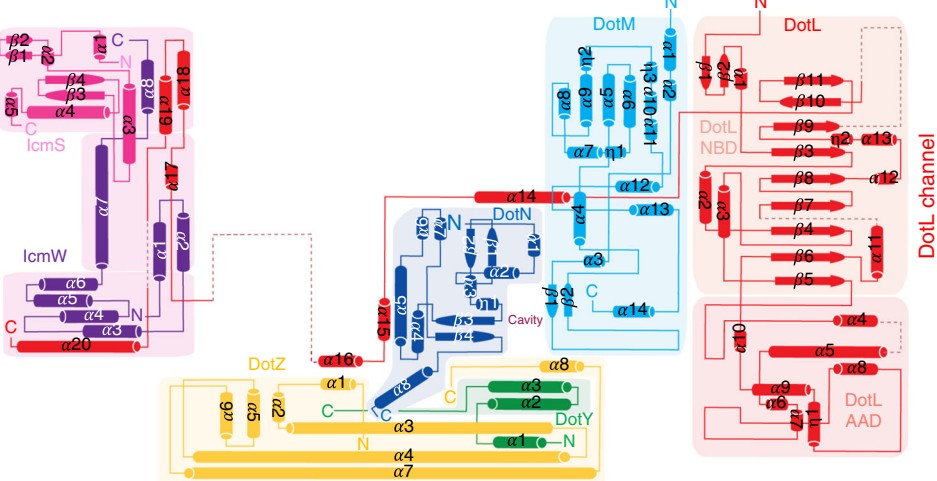

**Fig. 2 Topology diagram of the T4CC hetero-pentameric core structure.** Color-coding for DotL, DotM, DotN, DotZ, and DotY is as in Fig. 1e. IcmS and IcmW are in magenta and purple, respectively. Strands and helices are represented as arrows and cylinders, respectively. Secondary structures are labelled, as well as N- and C-termini. For DotL, NBD and AAD stands for nucleotide-binding and all alpha-helical domains, respectively. The location of the DotL channel in the hexamer and the cavity in which the DotM-bound Glu-rich SP inserts are indicated.

**Validation of the T4CC hetero-pentameric structure.** In order to validate the structure of the T4CC hetero-pentameric core, six regions involved in interfaces between two proteins within the complex structure were targeted for mutations (named M1–M6). These regions were chosen because of the multiplicity of contacts that they make. The location and structure of these regions are shown in Fig. 4a. M1 comprised the mutations T205R, L208R, Y211R in DotM α4α5 loop at the interface of DotM and DotN. For M2, the C-terminal region from residues 200 to the C-terminus of DotN corresponding to the C-terminal half of α8, which makes contact within the wedge of DotZ, were deleted. Another deletion mutant, M3, removed DotZ residues from residue 283 (just after α8) to the C-terminus. This region interacts with α8 of DotN and β1β2 of DotM. Two mutants, M4 (Q326R, T327R) and M5 (A363R, E364R, D366R), in DotM and DotL, respectively, were obtained, each affecting the contacts between residues in α11α12 in DotM and residues in α9α10 of DotL. Finally, another mutant (M6) was obtained by mutating another contact region between DotM and DotL at residues V300R, P302R, and S303R of the DotM η2η3 loop. All mutations were introduced as described in Methods. All mutants were tested for their ability to grow intracellularly in *A. castellanii* as described above for the *dotY* and *dotZ* deletion mutants (Table 1). All mutations introduced at the DotZ-DotN interface inhibited growth to the same extent as the deletion of the entire *dotZ* gene (M2, M3). Mutations at the interface of DotM and DotL (M4, M5, and M6) or of DotN and DotM (M1) resulted in total growth inhibition (Table 1). We conclude that the interfaces observed in the T4CC core structure provide an accurate account of the interactions taking place in vivo within the complex.

**The T4CC as a hexamer of hetero-pentameric units.** DotL belongs to the VirD4-family of AAA + ATPases which may purify as monomers but all function as hexamers[18]. Moreover, the only known structure of a representative VirD4 protein is that of TrwB, encoded by the R388 T4S system, and this structure is hexameric[19]. Thus, we used the TrwB structure to model a hexamer of DotL and consequently a hexamer of the *Legionella* T4CC hetero-pentameric core (Fig. 5). The resulting structure resembles a starfish, 26 nm in diameter (Fig. 5a). DotL forms a channel with an inverted funnel shape, constricted (2.0 nm) at the

base and flaring up to 4.3 nm near the membrane (Fig. 5a). Preceding the DotL channel lays a 6.7 nm diameter chamber, wide open on the cytosolic side but only accessible through the 2 nm DotL constriction on the membrane side (Fig. 5a). The interface between adjacent DotL molecules is extensive with circa 2,000 Å$^2$ of buried surface area in each DotL subunit. It is essentially similar to that of TrwB and therefore will not be described here (see however Supplementary Figs. 3f and 4j, k for details). A published mutational study of DotL[11] identified 12 residues across the DotL sequence, which, when mutated, resulted in intracellular growth defects. Three of these residues locate at the proposed DotL-DotL interface in the T4CC hexamer (Fig. 4b). These mutants provide validation for the proposed interface involved in hexamer formation, indicating its functional relevance in vivo. In the T4CC hexameric form, the membrane regions of both DotL and DotM would be expected to form a trans-membrane channel. Hexameric assembly might be induced by effector binding or a constitutive hexamer might be formed in vivo in the membrane environment. Kwak et al.[12] also proposed a hexameric model based on TrwB. Our model differs considerably. That's because the hexameric model proposed by Kwak et al. is based on multiple structures of separate subparts and some elements (DotM) were missing. In contrast, our model is based on the structure of an intact, fully assembled, complex. Differences include the following: the structure of DotL is here solved; DotM's position is experimentally established within the T4CC; IcmSW does not locate next to DotN but instead protrudes out; and we show that the T4CC actually include two additional proteins, DotY and DotZ.

**The IcmSW positional flexibility.** Single particle analysis of the T4CC revealed a U-shape density reminiscent of the structure of IcmSW bound to a DotL C-terminal peptide[12] (termed "IcmSW-DotL$_{672-783}$"; Supplementary Fig. 5a, b). Focusing on this U-shape density, we obtained a 9.7 Å resolution reconstruction into which the crystal structure of IcmSW-DotL$_{672-783}$ could easily be docked (correlation coefficient of 0.92; Supplementary Fig. 6). We concluded that the U-shape density that we observed does indeed correspond to IcmSW bound to DotL$_{672-783}$. LvgA, which co-purifies with the T4CC and is known to bind IcmSW[12], is absent from the structure determined here, likely because it is either too

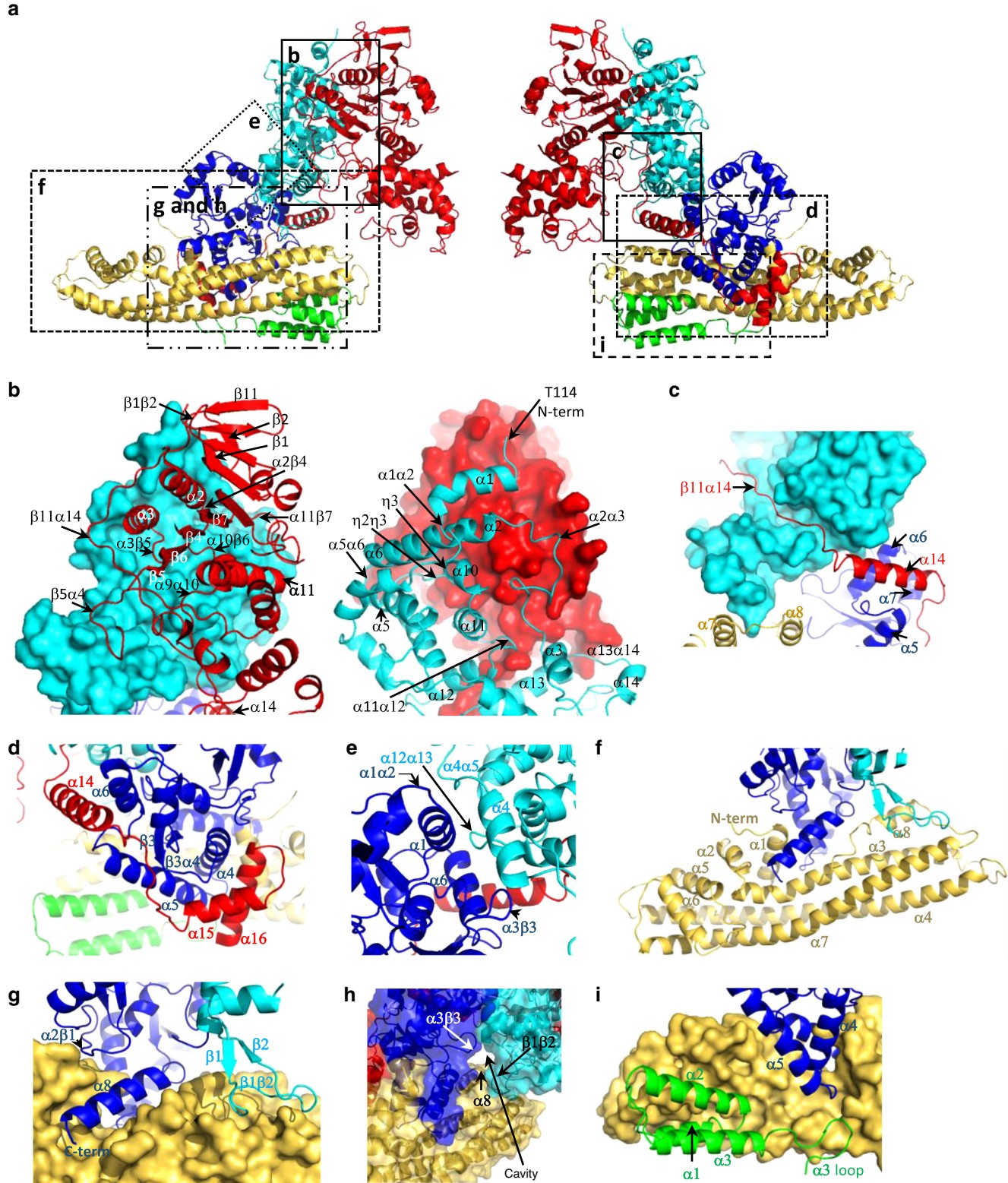

flexible or dissociates upon freezing during grid preparation, a well-known effect on protein complexes[20].

We next aimed to generate reconstructions of the entire T4CC including IcmSW. Three-dimensional (3D) classifications allowed us to resolve multiple orientations of the IcmSW module relative to the hetero-pentameric core (Fig. 5b, c and Supplementary Fig. 5c). IcmSW's positional flexibility is afforded by the DotL residues immediately preceding the IcmSW-binding region. In

the context of the hexamer model, maps superposition shows that the IcmSW module swings at the base of the structure in a trajectory that directs the module in and out of the DotL channel (Fig. 5b, c).

**Acidic Glu-rich SP effector binding**. As mentioned before, one subset of effectors does not rely on IcmSW for transport, but

**Fig. 3 Assembly of the T4CC. a** Locations of the various protein-protein interfaces shown in (**b-i**). In all panels, secondary structures involved in interactions are labelled. **b** The DotL-DotM interface. At left, DotL and DotM are shown in red ribbon and cyan surface, respectively. At right, DotM and DotL are shown in cyan ribbon and red surface, respectively. **c** The DotL-DotM interface (continued). This view focuses on the β11α14 part of the DotL tail (in red ribbon) which here interacts with a groove of DotM (in cyan surface) between two of its domains. **d** Interaction between the α14 to α16 part of the DotL tail (in red ribbon) with DotN (in blue ribbon). **e** The DotM-DotN interface. DotM and DotN are shown in cyan and blue ribbon, respectively. **f** Structure of DotZ (orange yellow ribbon) and its interface with DotN (blue ribbon) and DotM (cyan ribbon). All secondary structures in DotZ are labelled as well as its N- and C-terminus. **g** Details of the interface between DotZ (in orange yellow surface) and DotN (in blue ribbon) and DotM (in cyan ribbon). A groove between α1 and α3 is clearly visible in DotZ, into which α8 of DotN inserts. DotM makes contact with DotZ via the long β1β2 loop. **h** β1β2 of DotM together with α3β3 of DotN and α8 of DotZ form a cavity located just below the Glu-rich effector signal peptide binding site on DotM. **i** Interface between DotY (in green ribbon), DotN (in blue ribbon) and DotZ (in orange yellow surface).

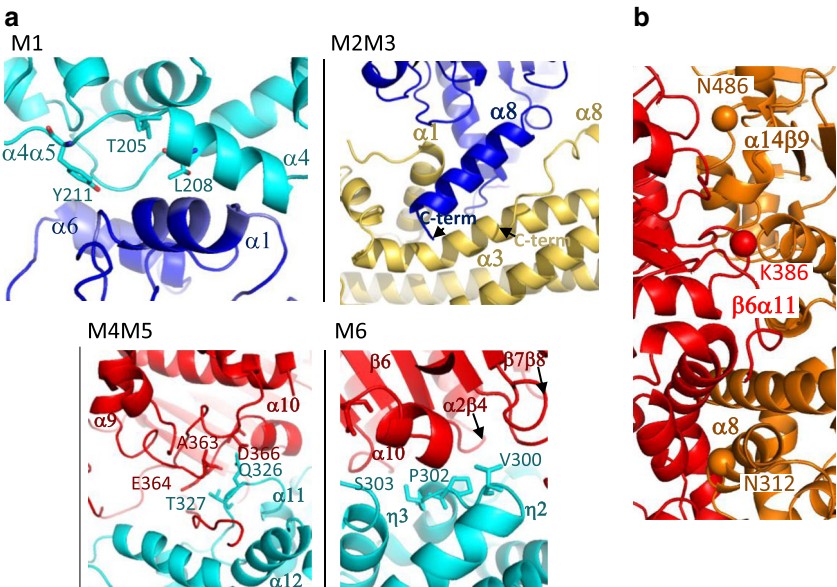

**Fig. 4 Validation of the hetero-pentameric T4CC complex and of the interface between DotL subunits in the proposed T4CC hexameric assembly. a** Location and structure of the regions M1 to M6 targeted for mutations. Mutated residues in each of the mutants are labelled as well as the secondary structures in which they are involved. Description of the 6 regions and mutants are provided in main text. **b** The interface between two adjacent DotL subunits in the proposed T4CC hexamer (shown in red and orange ribbons). Three mutations reported by Sutherland et al.[11] to affect intracellular growth locate to this interface. The Cα atom of these residues is shown as a sphere. The residues as well as the secondary structures they belong to are labelled.

**Table 1 Probing the observed protein-protein interaction network by site-directed mutagenesis.**

|     | Protein | Interface | Mutated residues | Mutant in Lp01 WT | Mutant in Lp01 ΔB + pDotB | AC Intracellular growth (48 hr) |
|-----|---------|-----------|------------------|-------------------|----------------------------|----------------------------------|
| M1  | DotM    | M-N       | T205R, L208R, Y211R | ✗ | ✓ | No growth |
| M2  | DotN    | N-DotZ    | 200-end deletion | ✓ |   | 27(±16)% of WT |
| M3  | DotZ    | DotZ-N, M | 283-end deletion | ✓ |   | 30(±7)% of WT |
| M4  | DotM    | M-L       | Q326, T327R      | ✗ | ✓ | No growth |
| M5  | DotL    | M-L       | A363R, E364R, D366R | ✗ | ✓ | No growth |
| M6  | DotM    | M-L       | V300R, P302R, S303R | ✗ | ✓ | No growth |

Six mutants (M1–M6) were generated targeting six different regions (shown in Fig. 4a) of the complex interfaces as described in the main text. Mutants were tested for their ability to grow intracellularly in *Acanthamoeba castellanii* (AC) as described in Methods. Data are presented as mean values +/− standard deviation. Source data are provided as a Source Data file.

instead binds directly to DotM[14]. These effectors are characterised by a particularly acidic Glu-rich SP. Previously, we characterised the surface of DotM involved in binding of Glu-rich SP and generated a structural model of the DotM-SP interaction[14]. When superposing this model onto the T4CC hetero-pentameric core structure using the DotM structures in both (Fig. 5d), the N-terminal end of the Glu-rich SP is observed inserting within the cavity mentioned above formed between DotM, DotZ, and DotN (Figs. 2, 3h and 5d). In the context of the T4CC hexamer (Fig. 5d), it can be seen that, by going through the cavity, the peptide reaches out to the DotL channel.

**Models for IcmSW- and DotM-dependent transport**. The results presented here provide mechanistic models for recruitment and delivery of two types of effectors, the IcmSW-dependent class and the acidic Glu-rich SP one. These models are detailed in Fig. 6a, b. The positional flexibility that we observe for IcmSW would provide a means for this module to scan the environment and maximise effectors capture while the defined trajectory we observe will facilitate their delivery to the DotL channel. On the other hand, Glu-rich SP-containing effectors bind to a region of DotM that induces the SP to insert into a cavity formed by three of the T4CC proteins. For the full-length

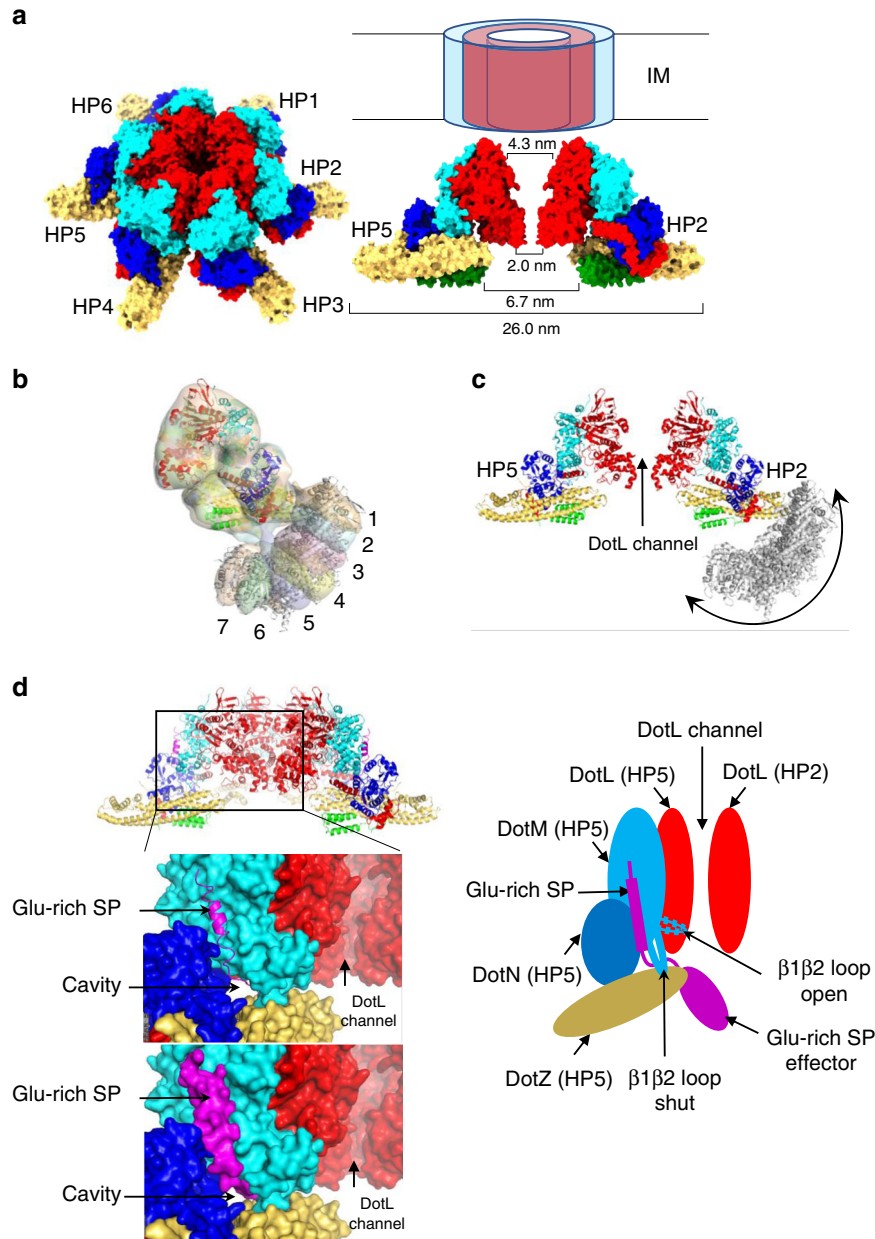

**Fig. 5 The T4CC hexamer and effectors recruitment and trajectory. a** Tilted and side views of the T4CC hexamer. "HP" refers to the hetero-pentameric unit of the T4CC core, 6 units of which form the T4CC hexamer. For the side view, only two hetero-pentameric core units are shown (HP2 and 5). The dimensions of the various parts of the T4CC channel are indicated. Proteins are shown in surface representations color-coded as in Fig. 1e. The trans-membrane segments of DotM and DotL are shown schematically as cylinders inserted through the IM. No structural information is yet available for this channel. **b** Superposition of 7 maps of the T4CC including IcmSW. The map superposition reports on 7 different positions for the IcmSW module. Map details are reported in Supplementary Fig. 5c. **c** Superposition of the various IcmSW module structures (in grey ribbon) derived from the maps shown in b in the context of the T4CC hexamer. Two opposite hetero-pentameric units (HP2 and 5) are shown in ribbon color-coded as in Fig. 1e, but only one (HP2) reports on the various IcmSW positions that we observe. The double arrow indicates the trajectory of the IcmSW module, which brings the module in and out of the DotL channel. **d** Access of Glu-rich SP-containing effectors to the DotL channel. Left, upper panel: location of the region of the T4CC hexamer shown in the two lower panels. Left, lower panels: the surfaces of DotM (cyan), DotN (blue) and DotZ (orange yellow) join up to form a cavity into which the Glu-rich SP bound to DotM (shown in magenta ribbon (upper panel) or surface (lower panel)) inserts. Four DotL subunits are shown in red surface, providing view of the channel within. At right, schematic representation of the view at left, except for the rest of the Glu-rich SP-containing effector being represented at the N-terminus of the Glu-rich SP. The β1β2 loop is labelled and is shut i.e. observed making interactions with the surface of DotZ. For the Glu-rich SP effector bound to DotM (through the Glu-rich SP shown in magenta) to insert within the DotL channel, the loop would need to open (labelled "open").

**Fig. 6 Mechanistic models of substrate recruitment and delivery by the *Legionella* T4CC. a** IcmSW-dependent effectors. Positional flexibility of the IcmSW module relative to the rest of the structure provides a means to scan the immediate environment to maximise substrate capture (step 1) by the T4CC. Once bound, because of the motions trajectory, the effector will be delivered to the DotL channel (step 2 "delivery"). Binding to IcmSW may induce IcmSW-dependent effectors to partially unfold[23]. However, structural details of effector-IcmSW interactions are not known. Additionally, DotL might be able to use its ATPase activity for the dual purpose of unfolding and transport (step 3 "translocation"). LvgA is not shown because it is not present in the structure presented here. **b** Glu-rich SP-containing effectors. Glu-rich SP-containing effectors bind to a region of DotM that induces the SP to insert into a cavity formed by three of the T4CC proteins. For the full-length effector protein to bind there, there would be a requirement for the cavity to open up (step 1 "capture"). To do so, we hypothesize that the β1β2 loop of DotM might swing out, allowing the DotM-bound effector to slot in to position itself under the DotL channel (step 2 "delivery"). How would then the effector in the DotL channel unfold remains unclear but it cannot be excluded that DotL might use its ATP-driven power to mediate unfolding (step 3 "translocation").

effector protein to pass through this cavity, the cavity would need to open up. Only β1β2 of DotM can swing out because (1) it is the only structural element mounted on a flexible linker, and (2) it is weakly anchored to DotZ (Fig. 3g). Once the DotM-bound effector has slotted into place through the open cavity, it would lie under the DotL channel, ideally positioned for translocation. Further steps beyond binding and delivery may include partial unfolding[21] by either IcmSW for IcmSW-dependent effectors in a way reminiscent of type III secretion chaperones[22,23] or by DotL itself for other effectors.

The structure of the T4CC from *L. pneumophila* reveals a remarkably versatile, multi-site, recruitment and delivery platform. Given the large number of effectors *Legionella* is able to secrete[3,24], it is likely that other capturing and delivery mechanisms involving the T4CC will be unravelled in the future. Such a large structure may indeed contain additional sites for binding of other types of effectors which are neither IcmSW dependent nor dependent on DotM. Interestingly, the T4S system is not the only secretion systems endowed with a large platform for effector recruitment and delivery[25]. Thus, our structural investigation might provide a potential paradigm on effector recruitment by other secretion machines. Finally, it could be argued that such a multi-site effector-binding platform might provide scope for temporal regulation of effector secretion, some binding sites functioning in early stages of secretion while others coming in later.

## Methods

**Bacterial strains and constructs**. Strains, plasmids and oligonucleotides used in this study are shown in Supplementary Tables 1, 2 and 3, respectively.

Isogenic Lp02 strains were produced as previously described[26]. To generate the DotL-Strep tag construct, *dotL* was first cloned into suicide plasmid pSR47S with 1000 bp upstream and downstream of the gene's 5′ and 3′ sequences. Then, the sequence encoding a Strep-tag (SASWSHPQFEK) was introduced to the 3′-end of *dotL*.

For production of the KO strains Δlpg0294 (DotY) and Δlpg1549 (DotZ) in the Lp01 background, genes were cloned with 1000 bp upstream and downstream to pSR47S, later deleted leaving a double stop codon TAA after the first ATG,

followed by 20–25 bp of the gene's 5′ and 3′ sequences. For production of the double knockout strain, after creation of Δlpg0294 strain, additional mutagenesis was performed with the Δlpg1549 construct. All strains were verified by colony PCR.

For DotY/DotZ complementation assays, *dotY* and *dotZ* were cloned into pJB1806 plasmid using InFusion. Wild-type *dotY* and *dotZ* were cloned into the pJB1806 backbone with 200 bp upstream and downstream, so that their native promotor is used for expression.

Interface mutations were introduced to the pSR47S constructs of *dotL*, *dotM*, *DotN*, and *dotZ* by In-Fusion or by Quick change. First attempt of mutagenesis was conducted in Lp01 wild-type background, and in cases of no positive hits (0/70 colonies), strains were generated in the background of Lp01 Δ*dotB*, a strain with inactive T4BSS. All mutated strains were verified by colony-PCR, followed by sequencing of the mutated region.

For DotB complementation assays, *dotB* fused at its 3′-end to the Strep-tag encoding sequence mentioned above was cloned into pMMB207 plasmid using InFusion.

For translocation assays, the sequences encoding DotY, DotZ, effector Lem21 and effector Lpg1663 were cloned at the 3′-end of the Cya gene in the pMMB207 background[27,28]. Cya-RalF, Cya-CegC3 and Cya-LegC8 were reported previously[14,28].

**Sample purification**. *Legionella* cells were grown on charcoal yeast extract (CYE) plates or AYE medium containing appropriate antibiotics (100 μg ml$^{-1}$ streptomycin and 10 μg ml$^{-1}$ chloramphenicol) as previously described[28].

For DotL$_{Strep}$ purification, 48 h heavy patch cells were inoculated and grown for additional 26 h in AYE medium and supplements to achieve a final OD$_{600}$ of 3.2–3.6. Cells were harvested and resuspended in buffer LPA (50 mM Tris pH 8.0, 0.2 M NaCl, 2 mM EDTA, 20 mM MgSO4) and 0.5 M sucrose, 0.1 mg/ml lysozyme, DNAse I and protease inhibitor (PI) (Roche). After rotation for 45 min at 4 °C, cells were spun down and then re-suspended in buffer LPB (50 mM Tris pH 8.0, 2 mM EDTA, 20 mM MgSO4, and PI), followed by 3 rounds of high pressure (40,000 psi) homogenisation. The lysate was centrifuged at 17,300 × g for 20 min to remove the cell debris, followed by ultracentrifugation at 167,000 × g for 2 h. For each 1 gr of membranes, 6 ml of LPA buffer was added and membranes were frozen at −80 °C until needed. An additional 6 ml of re-suspension buffer LPA and 2.5% DDM (*n*-Dodecyl-β-D-Maltopyranoside, Anatrace) was added after thawing. Solubilized membranes were gently shaken at room temperature for 2 h, then centrifuged again at 142,000 g for 30 min to remove insoluble materials.

Soluble membranes were loaded on 5 ml StrepTrap column (GE Healthcare), followed by extensive wash in LPA buffer and 0.05 % DDM, and eluted in LPA buffer with 0.05% DDM and 2.5 mM desthiobiotin (Sigma). Eluted fractions were analysed by SDS-PAGE, pooled, concentrated and loaded on SEC column Superose 6 (GE Healthcare). Peak fractions were pooled, concentrated, and protein concentration was determined by OD$_{280}$ measurement. To remove DDM, the

concentrated complex solution was incubated with Amphipol A8-35 (Anatrace) at 1:5 ratio for 4 h, followed by overnight incubation with biobeads (Biorad). The sample was then reloaded on the Superose 6 column, and peak fractions were collected and concentrated for cryo-EM studies.

**Cryo-EM grid preparation and data acquisition**. Aliquots of the purified T4CC were applied to negatively glow discharged UltrAuFoil R1.2/1.3 grids (Quantifoil, Germany) and vitrified in liquid ethane using a Vitrobot Mark IV (Thermo Fisher, USA) at 4 °C and 94% humidity. The data were collected at the eBIC National facility (Diamond Light Source, UK) and ISMB Birkbeck EM facility using Titan Krios microscopes (Thermo Fisher, USA) operated at 300 keV and equipped with a Quantum energy filter. The images were collected with a post-GIF K2 Summit direct electron detector operating in counting mode, at a nominal magnification of 130,000, corresponding to a pixel size of 1.047 Å. An energy slit with a width of 20 eV was used during data collection. The dose rate on the specimen was set to 4.9 e per pixel per second, and a total dose of 54 e Å$^{-2}$ was fractionated over 48 frames. Data were collected using EPU software (Thermo Fisher, USA) with a nominal defocus range set from −1.5 μm to −3.5 μm. A total of 19,491 micrographs were collected.

**Cryo-EM data processing**. RELION 3.0[29] was used for motion correction and dose weighting with MOTIONCOR2[30] followed by CTF estimation using CTFFIND v4.1[31]. An initial low-resolution map was obtained using RELION 3.0 following the workflow described in Zivanov et al.[32]. Reprojections of this map were used to pick particles with GAUTOMATCH v0.56[33]. Dataset was subjected to multiple rounds of 2D and Ab-initio classifications with CRYOSPARC v0.6.5[34] leading to selection of 626,230 out 8,702,486 particles.

Selected particles were re-extracted from 16,861 micrographs using RELION 3.0, following by 3D refinement and 3D classification that resulted in further selection of 541,522 particles. These particles were re-centered, used for 3D refinement with a mask focusing on the DotLMNYZ density, and subjected to 3D classification with the same mask without image alignment using Tau = 20. The two best resulting classes corresponding to 241,838 particles were selected. To limit anisotropy and improve the quality of the map, ~20,000 particles corresponding to preferential views were removed from the star files using rlnMaxValueProbDistribution criteria. The final subset of 219,593 particles was imported to CRYOSPARC v2.9.0, to perform Non-Uniform Refinement that resulted in an electron density map with a nominal resolution of 3.7 Å as estimated using gold standard Fourier shell correlation (FSC) with a 0.143 threshold (Supplementary Fig. 1 and Supplementary Table 4). This map was AutoSharpen using PHENIX v1.14[35].

To determine the structure of the U-shaped domain and ascertain that it corresponds to IcmSW, we selected particles with the characteristic U-shape, either alone or attached to the T4CC core. Indeed, a small fraction of the U-shape density was observed detached from the T4CC core. The 541,522 particles set selected during the T4CC core structure determination (see above) were re-extracted using RELION 3.0 with a shift and centering on the U-shape feature. 3D classification was performed using parameter Tau = 2 without alignment, and 103,532 particles corresponding to 4 classes showing U-shape density were selected. In addition, a set of 64,750 particles with the characteristic U-shape were selected after 2D classification and ab-initio 3D classification using CRYOSPARC. These two sets of particles were then combined and subjected to 2D classification and ab-initio classification using CRYOSPARC v0.6.5. A final subset of 18,210 particles was selected and subjected to homogeneous refinement. The resulting 9.7 Å resolution map (as estimated by the gold standard FSC with a 0.143 threshold (Supplementary Fig. 6)) was sharpened using CRYOSPARC v0.6.5 with a B factor value of −1500. The DotL-Cter IcmSW crystal structure (PDB ID 5×1E) was docked as a rigid body into the final map using CHIMERA v1.13.1[36].

Different positions of IcmSW relative to the hetero-pentameric T4CC core were resolved using CRYOSPARC ab-initio classification with the high-resolution limited to 20 Å (Supplementary Fig. 5). Initially 10 classes were obtained using the subset of 541,522 particles selected during the T4CC core structure determination. Classes showing the IcmSW domain in the same position relative to the T4CC core were selected and combined. Ab-initio classification and selection of classes were repeated two more times. All maps showing a clear density for the IcmSW domain were aligned using the hetero-pentameric T4CC core region and CHIMERA v1.13.1. Seven maps were selected to represent the extent of IcmSW domain motion relative to the hetero-pentameric T4CC core (Fig. 5b, c).

**Model building and refinement**. I-TASSER[37] was used to generate a model of the DotL ATPase domain (100–589) derived from the TrwB structure (PDB entry 1GKI). This model was combined with the DotL part of the DotL$_{590-659}$-DotN crystal structure solved previously (PDB ID 5 × 42[12]) to generate the starting model of DotL$_{104-658}$. For DotM and DotN, the previous crystals structures by Meir et al.[14] and Kwak et al.[12] were used (PDB IDs 6EXD and 5×42, respectively). DotY and DotZ were built de novo in COOT v0.8.9.1[38] based on the density map and secondary structure prediction (PSIPRED 4.0[39]). Simulated annealing in the initial rounds of real-space refinement with PHENIX was used.

The entire structure of the T4CC hetero-pentameric core (DotLMNYZ) was improved by iterative rounds of manual adjustment in COOT v0.8.9.1 followed by real-space refinement in PHENIX v1.14 using secondary structure restraints. MOLPROBITY v4.4[40] was used to evaluate the quality of the structures. All data and model statistics are reported in Supplementary Table 4.

Interaction analysis was conducted using PISA server[41], and structure representations were generated using UCSF CHIMERA v1.13.1, CHIMERAX v0.91 and PYMOL v2.3.2[42].

**Cell Culture**. CHO FcγRII cells[43] used for translocation assays were cultured at 37 °C in 5% $CO_2$ in RPMI-1640 plus 10% FBS. *A. castellanii* (ATCC 30234) were cultured routinely at room temperature in ATCC medium 712 (PYG).

***Legionella* intracellular growth in eukaryotic hosts**. Intracellular growth assays were performed as previously described[44]. Specifically, *A. castellanii* were infected in AC medium. Cells were plated at $2 \times 10^5$ cells/ well and incubated at 37 °C 2 h prior infection. Two-day heavy patch bacterial strains were grown on CYE plates with appropriate antibiotics (100 μg ml$^{-1}$ streptomycin for WT and mutant strains, supplemented with 10 μg ml$^{-1}$ chloramphenicol for the strains containing the complementing plasmids). Bacterial strains were added to *A. castellanii* plates at MOI of 0.1 ($2 \times 10^4$ cells per well) followed by centrifugation for 5 min at $350 \times g$ at room temperature and incubation at 37 °C for 1 h. For intracellular growth of the interface mutants, *A. castellanii* were infected in AC medium containing 1 mM IPTG. A Δ*dotB* strain complemented by pMMB207:DotB with or without IPTG in the media was included as positive and negative control, respectively.

**CYA assay**. Cya assays were conducted as previously described[14]. Specifically, CHO FcγRII cells ($1 \times 10^5$ cells per well) were placed into 24-well tissue culture plates in α-MEM plus 10% FBS 1 day prior to infection. On the day of infection, 2-days heavy patch of *Legionella* strains (W.T., Δ*T4SS*, Δ*dotY*, Δ*dotZ*, or *dotYdotZ* double knockout mutants) transfected with the Cya-containing plasmids were diluted into α-MEM plus 10% FBS medium supplemented with Rabbit anti-*Legionella* antiserum diluted at a ratio of 1:1000 (which facilitates *Legionella* adhesion) and 0.5 mM IPTG (to induce Cya fusions), and incubated at R.T for half an hour prior to infection. The CHO FcγRII cell culture medium was aspirated before adding to each well the corresponding *Legionella* strains ($3.0 \times 10^6$ bacteria per well). The plates were centrifuged onto a confluent monolayer of host cells for 5 min at $200 \times g$. Plates were immediately warmed in a 37 °C water bath for 5 min, then placed in a $CO_2$ incubator for a total of 1 h. Cells were washed three times with ice-cold PBS and lysed in 200 μl of extraction solution (50 mN HCl/0.1% Triton X-100) on ice. After boiling for 5 min, extracts were neutralized with 12 μl of 0.5 M NaOH and cAMP was extracted with 2 volumes of ethanol. Insoluble materials were pelleted by centrifugation, and the soluble materials containing cAMP were lyophilized. The cAMP levels were determined for each extract by using an ELISA kit according to manufacturer's instructions (Amersham Biosciences, RPN-225).

**Statistical analysis**. Statistical analysis was performed with GraphPad Prism v.5.0 (GraphPad Software, La Jolla, CA, USA). For comparison of two groups, Lp01 WT against mutants, an unpaired *t* test was employed. A *P* value of <0.05 was considered statistically significant. All experiments were performed at least three times, each strain with three biological triplicates. The data are expressed as mean ± standard deviation (s.d.). For the Cya translocation assays, effectors translocation values in mutants were normalized against their ratio to WT. For the Cya effector translocation assay, we also employed one sample *t* test Vs. 1 to determine significance between WT and mutant strains. *P* values were calculated and still showed a significant difference.

**Reporting summary**. Further information on research design is available in the Nature Research Reporting Summary linked to this article.

## Data availability
Entry codes for the EM density map and the atomic model of the hetero-pentameric T4CC core are EMD-10350 and PDB ID 6SZ9, respectively [https://doi.org/10.2210/pdb6SZ9/pdb]. The source data underlying Fig. 1a–c, Table 1, and Supplementary Fig. 2 are provided as a Source Data file. Source data are provided with this paper.

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

## Acknowledgements

This work was funded by ERC grant 321630 and Wellcome grant 098302 to G.W. and NIAID grants R21AI130671 and R37AI041699 to CR. Most of the Cryo-EM data for this investigation were collected at the ISMB EM facility at Birkbeck College, University of London with financial support from Wellcome (202679/Z/16/Z and 206166/Z/17/Z). We would also like to thank Diamond Light Source for access to the cryo-EM facilities at the UK National electron bio-imaging centre (eBIC, proposal EM14704) funded by the Wellcome Trust, the Medical Research Council UK and the Biotechnology and Biological Sciences Research Council. We would like to thanks Dr. David Houldershaw for IT support and script writing, Professor Maya Topf for modeling, Dr. Nikos Pinotsis for advice in model building, and the members of the Waksman and Roy labs for scientific discussions.

## Author contributions

A.M. cloned, expressed and purified the T4CC. M.K.H. and A.R. prepared the EM grids and NL collected EM data. A.M. performed initial data processing. The bulk of EM processing for both high and low resolution works was carried out by K.M. and K.M. built and refined the DotLMNYZ and DotLMNYZ-IcmSW models. A.M. and D.C. generated the *Legionella* mutants and A.M. tested them. C.R. supervised the biological work. A.M. and G.W. supervised the biochemical work. G.W. supervised the structural work and wrote the article.

## Competing interests

The authors declare no competing interests.
