## [Peer Review File · Nature Communications]

Reviewers' comments:

Reviewer #1 (Remarks to the Author):

Meir and colleagues report the structure and substrate recognition modes of the Type IV coupling complex (T4CC) from *Legionella pneumophila*, which functions to export effector proteins into host cells. The authors have purified the complex and discovered two previously unknown proteins that form the T4CC, namely DotY and DotZ, and determined that these play significant role in translocation of effector proteins. Using cryo-electron microscopy, the authors have obtained a model of the entire complex consisting of core proteins ATPase Dot L, Dot M/N and Dot Y/Z with and without the module proteins IcmS and IcmW. The cryoEM structure is resolved with average 3.7 angstrom resolution, which is sufficient to separate DotL, DotM, DotN, DotZ and partially DotY structures, excluding transmembrane parts. The authors also have data that illuminate two separate modes of effector transport – one is IcmSW dependant, where flexible IcmSW chaperone helps the effector protein get inside the channel, and the other in which Glu-rich signal peptides bind to DotM, and are then inserted into the channel. The authors suggest that the latter requires conformational change of DotM. Combining their results, a plausible model is proposed to explain why IcmSW independent effectors require an “E-block motif/Glu rich repeat” in their C-terminal signal sequence for export.

Overall, the experiments are well-validated and findings provide important biological insight. Determining structure and the mechanism of action of DotLMNYZ/IcmSW may serve as a very important tool in understanding type IV secretion system, especially in human pathogens and help us solve how *Legionella* manages to survive in intercellular environment.

General comments

The manuscript can be considerably improved by explaining why their model differs from the one built by Kwak et al (2017), who also used the related TrwB hexameric structure to fit their crystal structures; the reader is left to assume that the addition of unexpected structures Dot Y (Ipg0294) and Dot Z (Ipg1549) led to a different model than what was expected based on the previous separate crystal structures docked into TrwB. In addition, no attempt was made to determine where LvgA fits into their model (apart from mentioning that it was not captured in their complex), and how it affects effector transport (presumably the LvgA-LcmSW interaction still occurs in similar manner to found in crystal structure?).

It would be helpful if the authors could explain more clearly how the identities of Dot Y and Z were found (mass spec analyses of purified SDS-PAGE bands?).

The interfaces between the subunits in Extended Fig 3 and rationale of choosing the mutations is not so clear (appear to also use multiple sequence alignment to find conserved amino acids in interface regions?).

The transport model proposed in Figure 4 is quite speculative considering the following points: 1) it is not known how the effector protein would interact with IcmSW, and if LvgA is required, 2) the authors did not state how they arrived at a model of the Glu-rich signal peptide interacting with the T4CC (page 8 lines 3-4), 3) it is not known if the effector is unfolded upon entering the Dot L channel, which they imply in the model, and 4) it is not known if the effector interacts with the hexameric assembly or if interaction induces oligomerisation. It would be good if the authors could address some of these issues.

Authors could comment more (if possible) on what happens to effector proteins during translocation, what is (if possible) structural similarity between them, and what was the reasoning

behind picking these ones to test. What draws attention is lack of structural representations of movement (swing) of DotM beta-1beta-2 loop, which would clarify authors' speculations about this exact mechanism.

It would also be helpful if the authors had other supporting data from the proposed hexameric organisation.

Technical Improvements

1) Statistical analyses for CYA assay (Figures 1b/c) needs improvement. An unpaired t-test requires two assumptions: 1) that the data follows a normal distribution, and 2) the variances between the compared groups are equal. The authors either need to conduct a test for normality if the sample size is large enough (e.g Shapiro-Wilk) or have a compelling reason to believe the distribution is normal. The authors should also test if the population variances of compared groups are equal, or use a different test where this assumption is not required (e.g Satterthwaite's approximate t-test). Also for this measurement there appears to be some significant variable background in the cAMP measurement (as there is signal for $\Delta T4SS$); the authors could conduct background subtraction using this average value to normalise values. Finally, since multiple comparisons are being made it would be a good idea to correct the p-value to take this into account.

2) There is a "bump" in their FSC which is likely caused by the transmembrane domain but it would be good to have some confidence that no dependence correlations were accidentally introduced at some point in the procedure.

3) It would be good to show some of the 3D classes. The 2D classes show some partially averaged out density; it is presumed this was the reason why a soft-edge masked was used for 3D refinement?

minor corrections:

Methods p2 L8 'pooled'?

Reviewer #2 (Remarks to the Author):

Overview

A decade after the publication of the first plasmid conjugation T4SS OM core complexes cryoEM/RXC hybrid structures (Chandran et al., 2009; Fronzes et al., 2009 then Low et al., 2014) and five years after that of a conjugative T4SS IM complex cryoEM/RXC hybrid structure, Prof. Waksman's team -often the leading group in this field- presents the first cryoEM/RXC hybrid structure of the cytoplasmic complex (CC) from the virulence protein translocation T4SS of Legionella. This is composed of an AAA+ ATPase (DotL), the substrate signal peptide binding-DotM, DotN, the substrate chaperones IcmSW and two new proteins shown here to be essential for T4SS function: DotY and DotZ.

Given the very high number of particles used (>200000 in the final structure), its only near-atomic resolution (~3.7 Å) must be due to its need to interact with the cytoplasmic membrane (Fig 1e)

and to its intrinsic flexibility (Fig. 3cd). Nevertheless, the dockings of available crystal structures (from this and other groups) reassuringly show excellent fits (extended data Fig. 1f).

The structure of the CC is interesting and novel in that it :

1) is likely hexameric (as are all AAA+ ATPases) in symmetry while the OMC is a 14-mer and the IMC is a 12-mer. Strangely however, this symmetry mismatch is never discussed (or at least acknowledged) in the manuscript.

2) through the localisation of known interaction partners, offers insight into the localisation of binding sites of both chaperoned and unchaperoned substrates.

3) indicates/displays localised conformational flexibility suggestive of mechanistic transport paths through the complex.

The last two points in particular are in my view highly likely to influence future experiments in this field for the foreseeable future.

Specific points:

a) Page 3, lines 17-18: it is not clear to me what deductive steps led the authors from a MW of 300 kDa in SEC-MALS to the conclusion that there is 1 copy of each protein in the complex? The measurement is approximative and at a superficial glance there seems to me to exist more than one solution to that equation. Isn't it rather that given the crystal structures and co-crystal structures they had available, the dockings strongly suggested this stoichiometry that also added up to about 300 kDa? Please would the authors clarify this?

b) Extended data Fig 3b, 2cd paragraph p6: while I acknowledge that structural biologists often resort to such experiments, I think that seeking to validate subunit interfaces using mutations predicted to disrupt them and hence lead to loss-of function-is flawed logic. Indeed as the author themselves discuss (reference 11) on the very next page: mutational studies very often lead to loss-of-function without necessarily affecting interfaces (in fact, in 9 out of 12 times in the case described here, apparently). Therefore, in my opinion, a trickier but much more elegant and convincing way to go (in the future, since here I think other lines of evidence independently support the dockings made) would be to try to make gain-of-function mutants allowing subunit-subunit interface stabilisation or crosslinking.

c) Fig 2c: did you check that the Cya-DotY and Cya-DotZ complement their respective single KO mutants? As they are essential to T4SS function, the fact that they are not translocated might otherwise only mean that they are simply non-functional or non-exportable (same logic as point d).

d) Extended data 4c: to allow easy comparison between the structures shown, please would you give numbers of particles used, resolution and dockings fit scores in the map Table as well?

e) Fig 3c (versus d): puzzled that there seems to be clear unfilled densities at the interfaces of IcmSW with the rest of the complexes for positions 4-7. This suggests to me that the DotZ helix bundle/DotL C-terminal extension must be able to swing quite far to the left, towards the mouth of the DotL channel. Did the authors think this too speculative to mention?

f) Extended Fig 5: despite the >18000 particles used in the reconstruction this subcomplex is poorly resolved. As it is not clear to me what additional point the authors are seeking to make by showing it, I would suggest clarifying this or deleting the figure to end the paper on a high note.

Minor points

-Page 8: for clarity, consider moving citation of ref. 14 from the end of the first (line 2) to end of the third sentence (line 4).

Reviewer #3 (Remarks to the Author):

Meir et al purified and characterized a key coupling complex, T4CC, for the type IV secretion system. The intact complex is formed by 6 proteins, including two previously uncharacterized ones. The main method used is single particle CryoEM and the reconstructed density map is of

sufficient quality to build atomic models for the hetero-pentameric core. The work is novel and of significance to the study of secretion systems.

The cryoEM and mutagenesis work for the purified T4CC complex are in general solid. My main concern is on the 1.6MD hexameric complex (hexamer of the actually solved structure in Fig 2) they presented in Figure 3. They did not provide any experimental evidences to show the existence of such a hexamer. If they can find a way to assemble and stabilize the proposed hexamer and image it directly, it will validate devoting one entire main figure as they presented in Figure 3 and the related portion of main text. Otherwise, the entire Figure 3 and related descriptions/discussions are basically speculations. Then they should tone down the discussion in the main text and possibly adjust related figures.

A second concern is Extended data figure 1e. There is a dip in the FSC curves between 7.9 and 5.2 angstrom. This is potentially a warning sign that something is not quite right in either the data processing or collection. The authors should investigate the cause of this.

My recommendation is for publication after revision.

Reviewer #1 (Remarks to the Author):

Comment 1: “The manuscript can be considerably improved by explaining why their model differs from the one built by Kwak et al (2017), who also used the related TrwB hexameric structure to fit their crystal structures; the reader is left to assume that the addition of unexpected structures Dot Y (Ipg0294) and Dot Z (Ipg1549) led to a different model than what was expected based on the previous separate crystal structures docked into TrwB.”

Response:

We apologise for not being clear enough. Although the hexameric structure was derived from TrwB in both our study and that of Kwak et al., the major difference is that Kwak et al. did not solve the structure of the intact hetero-trimeric DotLMN complex. Although their study is remarkable and represented a breakthrough at the time, they only solved structures of subparts. The model proposed by Kwak et al. is thus based on fragment structures that were put together without any experimental evidence because they didn't solve the structure of the intact complex. Thus, the relative positions of the fragment crystal structures was not obtained. In contrast, because we solve the structure of the intact hetero-pentameric complex that includes the DotLMN hetero-trimer, we know exactly the relative orientation of the complex's constituent parts. Thus, our structure provides an accurate structural and molecular description of the DotLMN complex.

Unfortunately, Kwak et al. have not released the coordinates of their hexameric complex but from Figure 5 of Kwak et al. we can provide an accurate evaluation of the differences of the model they proposed and our experimentally-derived structure:

- 1- Kwak et al does not provide an experimentally-derived structure of DotL. Our study does.
- 2- DotM is not positioned in the model proposed by Kwak et al. Our structure identifies the position and structure of DotM within the complex.
- 3- Kwak et al. positions IcmSW on the membrane-distal side of the complex just under DotM. Our structure definitely demonstrates that this is not the case. We experimentally derive the position of IcmSW relative to the DotLMN complex and it is not where Kwak et al. propose it might be.
- 4- In addition, we found two additional proteins bound, DotY and DotZ, two proteins Kwak et al. could not describe because they didn't seek to purify the DotLMN complex from Legionella cells. Instead they cloned the genes of interest (DotLMN and IcmSW) for expression in E. coli.

We thank the reviewer for pointing out our insufficient description of what makes our structure truly remarkable compared to that of Kwak et al. Because the coordinates of the model proposed by Kwak et al. are not available, we could not generate a figure comparing their model to ours. But we have added a note on page 9: “Kwak et al. (2017)¹² also proposed a hexameric model based on TrwB. Our model differs considerably. That's because the hexameric model proposed by Kwak et al. (2017) is based on multiple structures of separate subparts and some elements (DotM) were missing. In contrast, our model is based on the structure of an intact, fully-assembled, complex. Differences include the following: the structure of DotL is here solved; DotM's position is experimentally established within the T4CC; IcmSW does not locate next to DotN but instead protrudes out; and we show that the T4CC actually include two additional proteins, DotY and DotZ.”

Comment 2: “In addition, no attempt was made to determine where LvgA fits into their model (apart from mentioning that it was not captured in their complex), and how it affects effector transport (presumably the LvgA-IcmSW interaction still occurs in similar manner to found in crystal structure?)”.

Response:

LvgA co-purifies with the complex. However, density for LvgA was not found indicating that it is either disordered in our complex and therefore no density can be found for it, OR it has dissociated from IcmSW during freezing of the samples during grid preparation. It is not uncommon that complexes or

parts of a complex fall apart upon freezing during grid preparation. It is well known that the air-water interface formed during cryo-EM grid making can be a hostile environment for proteins (see for example Glaeser and Han (2017) *Biophys Rep* 3:1-7), leading to local complex dissociation and sometimes denaturation/unfolding of some complex components. We think this is what has happened with LvgA: it co-purifies but it dissociates during grid preparation upon freezing. We have modified the text accordingly on page 9: “LvgA, which co-purifies with the T4CC and is known to bind IcmSW¹², is absent from the structure determined here, likely because it is either too flexible or dissociates upon freezing during grid preparation, a well known effect on protein complexes¹⁸”.

Comment 3: “It would be helpful if the authors could explain more clearly how the identities of Dot Y and Z were found (mass spec analyses of purified SDS-PAGE bands?).”

Response: This was added in the Figure 1 legend: “Protein bands were identified by mass spectrometry and are labelled accordingly.”

Comment 4: “The interfaces between the subunits in Extended Fig 3 and rationale of choosing the mutations is not so clear (appear to also use multiple sequence alignment to find conserved amino acids in interface regions?).”

Response: we apologise for the lack of clarity. Although a sequence alignment is presented in Extended Data, it was not used in support of our mutational study. Our mutational study is based uniquely on inspection of the structure and identifying interface regions that we thought should be targeted for mutations. We have expanded the rationale and hope to have made it clearer by writing on page 7: “In order to validate the structure of the T4CC hetero-pentameric core, six regions involved in interfaces between two proteins within the complex structure were targeted for mutations (named M1-M6). These regions were chosen because of the multiplicity of contacts that they make.”

Comment 5: “The transport model proposed in Figure 4 is quite speculative considering the following points: 1) it is not known how the effector protein would interact with IcmSW, and if LvgA is required, 2) the authors did not state how they arrived at a model of the Glu-rich signal peptide interacting with the T4CC (page 8 lines 3-4), 3) it is not known if the effector is unfolded upon entering the Dot L channel, which they imply in the model, and 4) it is not known if the effector interacts with the hexameric assembly or if interaction induces oligomerisation. It would be good if the authors could address some of these issues.”

Response: We thank the reviewer for pointing out some of the more speculative elements of our model.

Response to point 1: It is indeed unknown how the effector protein would interact with IcmSW. But our model really does not say anything about this. It only shows schematically IcmSW interacting with an effector and indeed LvgA is not mentioned because it is not present in our structure. We have added in the legend to Figure 6 (formerly Figure 4): “However, structural details of effector-IcmSW interactions are not known” and “LvgA is not shown because it is not present in the structure presented here”.

Response to point 2: the model of the Glu-rich peptide bound to the T4CC is derived from the model of the Glu-rich peptide bound to DotM that we published in *Nature communications* in 2018. We have rephrased the description of what we’ve done by saying on page 10: “Previously, we characterised the surface of DotM involved in binding of Glu-rich SP and generated a structural model of the DotM-SP interaction. When superposing this model onto the T4CC hetero-pentameric core structure using the DotM structures in both (Fig. 5d), the N-terminal end of the Glu-rich SP is observed inserting within the cavity mentioned above formed between DotM, DotZ, and DotN”.

Response to point 3: One potential interpretation of the work by Amyot, W. M., deJesus, D. & Isberg, R. R. *Poison domains block transit of translocated substrates via the Legionella pneumophila*

Icm/Dot system. *Infect Immun* 81, 3239-3252, doi:10.1128/IAI.00552-13 (2013) is that indeed effectors require to be unfolded for translocation. But to comply with the reviewer's recommendation we have toned down this part of the paper. For example, on page 11, we now say "Further steps beyond binding and delivery may include partial unfolding..." or, on page 24, "Binding to IcmSW may induce IcmSW-dependent effectors to partially unfold..."

Response to point 4: it is indeed not known whether hexameric assembly is induced by effector binding or whether hexamerisation is constitutive *in vivo* in the membrane environment. We have added a sentence on page 9: "Hexameric assembly might be induced by effector binding or a constitutive hexamer might be formed *in vivo* in the membrane environment".

Comment 6: "Authors could comment more (if possible) on what happens to effector proteins during translocation, what is (if possible) structural similarity between them, and what was the reasoning behind picking these ones to test. What draws attention is lack of structural representations of movement (swing) of DotM beta-1beta-2 loop, which would clarify authors' speculations about this exact mechanism."

Response: We could indeed comment more on what happens to effector proteins during transport. But in the absence of additional data, it might be too speculative and therefore we would prefer not to expand on details of a mechanism for which evidence might still be lacking. In the schematic diagram in Figure 6, we show two positions for the beta1-beta2 loop. We could indeed speculate as to what the open position might look like structurally. But, in the absence of a structure of the open form of the loop, it would be speculative and, therefore, here again, we would prefer not to show details of an open form which must exist but for which we have no structure.

Comment 7: "It would also be helpful if the authors had other supporting data from the proposed hexameric organisation."

Response: It is not coincidental that when we generated a hexamer using TrwB, a well-known DotL homologue, the DotL-DotL interface thus modelled contained many of the residues Sutherland et al (2012) have described as affecting DotL activity. So making more mutations would not make our case stronger. However, as recommended by Reviewer 3, we have toned down considerably our proposed model. Please refer to our response to comment 1 of Reviewer 3 for more details.

Comment 8: "1) Statistical analyses for CYA assay (Figures 1b/c) needs improvement. An unpaired t-test requires two assumptions: 1) that the data follows a normal distribution, and 2) the variances between the compared groups are equal. The authors either need to conduct a test for normality if the sample size is large enough (e.g Shapiro-Wilk) or have a compelling reason to believe the distribution is normal. The authors should also test if the population variances of compared groups are equal, or use a different test where this assumption is not required (e.g Satterthwaite's approximate t-test). Also for this measurement there appears to be some significant variable background in the cAMP measurement (as there is signal for $\Delta T4SS$); the authors could conduct background subtraction using this average value to normalise values. Finally, since multiple comparisons are being made it would be a good idea to correct the p-value to take this into account."

Response: The Cya assay for effector translocation level measurements in *L. pneumophila*, along its unpaired t-test statistical analysis, has been first described in Nagai H et al. *Proc Natl Acad Sci U S A*. 2005 102(3):826-31. Since then, the assay, has been conducted and reported many times (e.g. Al-Khodor S et al *Mol Microbiol*. 200870(4):908-23, Amyot WM et al *Infect Immun*. 201381(9):3239-52, Lifshitz et al *Proc Natl Acad Sci U S A*. 201319;110(8):E707-15). Our data represents 3 independent experiments, each with triplicate repetitions. Due to the nature of the cAMP detection kit, the results presented are normalized to the cAMP values of the WT for each effector. Following the reviewer's comments, we consulted our statistical analysis unit in Yale University, and they advised us to check

our statistical treatment of the data using either the one sample t-test or the non-parametric U-test. Both tests confirm our initial results and therefore the figure was left as is.

Concerning the significant variable background in the cAMP measurement, this is not surprising: cells normally have intrinsic cAMP, usually around 10^2 - 10^3 fmol/well, hence the cAMP background in the Δ T4SS. cAMP levels between different effectors in the Δ T4SSS strain did not vary significantly.

Comment 9: “2) There is a “bump” in their FSC which is likely caused by the transmembrane domain but it would be good to have some confidence that no dependence correlations were accidentally introduced at some point in the procedure.”

Response: The bump in the FSC that we observed has been observed before for membrane proteins, likely due to the higher-than-usual proportion of disordered or semi-disordered regions. This has been the subject of a discussion on the CRYOSPARK bulletin board and the response by a specialist and confirmed by a programme user is that most (if not) all membrane proteins appear to display this bump in their FSC. Examples of structures that have the same bump in their FSC and were published in both general and specialist journals include: Basak S et al, 2018, Nature, 563: 270-274; Nguyen AH et al, 2019, NSMB, 26: 1123-1131; Lawrence RE et al, 2019, Science, 366: 971-977; Autzen HE et al, 2018, Science, 359: 228-232; or Chen Y et al, 2016, Science, 353: 887-891; to only cite a few. Others have mentioned a potential impact of beam tilt variations and this prompted us to refine this variable: this did not lead to the disappearance of the bump. Thus, the bump is likely a feature encountered in membrane protein structures determined by EM. However, the more compelling argument that our structure is correct is the following: in the hetero-pentameric DotLMNYZ complex, two structures were already known and determined using X-ray crystallography, that of DotM and that of DotN: in a blind exercise, we built the structure of these proteins de novo into the density and obtained the same structures. This demonstrates that there is no doubt that the data have been correctly processed, yielding electron density in which accurate models could be built.

Comment 10: “3) It would be good to show some of the 3D classes. The 2D classes show some partially averaged out density; it is presumed this was the reason why a soft-edge masked was used for 3D refinement?”

Response: We already present a number of 3D classes in Extended Figure 4c to show the positional flexibility of the IcmSW module. Concerning the use of a soft-edge mask, the reviewer is correct.

Comment 11: “minor corrections: Methods p2 L8 ‘pooled’?”

Response: corrected

Reviewer #2 (Remarks to the Author):

Comment 1: “a) Page 3, lines 17-18: it is not clear to me what deductive steps led the authors from a MW of 300 kDa in SEC-MALS to the conclusion that there is 1 copy of each protein in the complex? The measurement is approximative and at a superficial glance there seems to me to exist more than one solution to that equation. Isn't it rather that given the crystal structures and co-crystal structures they had available, the dockings strongly suggested this stoichiometry that also added up to about 300 kDa? Please would the authors clarify this?”

Response: The reviewer is absolutely right. We should have used a different wording. We have corrected and now say on page 4: “....consistent with a complex that may contain....”.

Comment 2: “b) Extended data Fig 3b, 2cd paragraph p6: while I acknowledge that structural biologists often resort to such experiments, I think that seeking to validate subunit interfaces using mutations predicted to disrupt them and hence lead to loss-of function-is flawed logic. Indeed as the author themselves discuss (reference 11) on the very next page: mutational studies very often lead to loss-of-function without necessarily affecting interfaces (in fact, in 9 out of 12 times in the case described here, apparently). Therefore, in my opinion, a trickier but much more elegant and convincing way to go (in the future, since here I think other lines of evidence independently support the dockings made) would be to try to make gain-of-function mutants allowing subunit-subunit interface stabilisation or crosslinking.”

Response: We appreciate what the reviewer is suggesting here. But, as he/she remarks, the strategy is standard practice and has shown its worth within the limits that the reviewer so beautifully outline. As a first line of validation, the strategy we have adopted is valid and widely accepted. In that context, we feel that finding gain-of-function mutations should be left for future work and for another paper.

Comment 3: “c) Fig 2c: did you check that the Cya-DotY and Cya-DotZ complement their respective single KO mutants? As they are essential to T4SS function, the fact that they are not translocated might otherwise only mean that they are simply non-functional or non-exportable (same logic as point d).”

Response: When testing for effector protein translocation using Cya fusion approach, assessment of the activity of the Cya-effector fusion construct is not required. The reason for this is that effectors are engaged by the secretion apparatus and translocated in an unfolded state, and this is typically mediated by recognition of a C-terminal secretion signal encoded by the effector, which is why the standard protocol is to fuse the Cya enzyme to the amino terminal position in the putative effector. Thus, as long as the fusion construct contains the signal sequence, the Cya enzyme should be delivered into the host cytosol. Because the Cya-DotY and Cya-DotZ proteins contain the entire coding sequence of the Dot protein, the C-terminal signal required for secretion is present.

Comment 4: “d) Extended data 4c: to allow easy comparison between the structures shown, please would you give numbers of particles used, resolution and dockings fit scores in the map Table as well?”

Response: We are a little confused by this comment as most of the numbers that the reviewer would like us to report are already reported, except for the resolution. We could indeed report the resolution but as indicated in the methods section, we limited the resolution to 20 Å.

Comment 5: “e) Fig 3c (versus d): puzzled that there seems to be clear unfilled densities at the interfaces of IcmSW with the rest of the complexes for positions 4-7. This suggests to me that the DotZ helix bundle/DotL C-terminal extension must be able to swing quite far to the left, towards the mouth of the DotL channel. Did the authors think this too speculative to mention?”

Response: We also noticed the unfilled densities in these maps. Our initial attempt at interpreting these densities were to assign them to the linker region of DotL between the IcmSW-binding site and the T4CC core-binding site. But the reviewer’s suggestion could also provide a possible interpretation. This illustrates the fact that there may be many potential interpretations, all rather too speculative to be mentioned.

Comment 6: “f) Extended Fig 5: despite the >18000 particles used in the reconstruction this subcomplex is poorly resolved. As it is not clear to me what additional point the authors are seeking to

make by showing it, I would suggest clarifying this or deleting the figure to end the paper on a high note.”

Response: We respectfully disagree with this suggestion. Early on during processing, we noticed the U-shape density and thought it could be ascribed to IcmSW, the crystal structure of which indeed shows that it has a U-shape. But we needed to ascertain that this U-shape density in our micrographs was indeed IcmSW. Extended Fig 5 provides the irrefutable proof that indeed the U-shape density is IcmSW. So it seems to us that Extended Fig 5 (now Supplementary Fig. 5) is important and must remain.

Comment 7: “Minor points -Page 8: for clarity, consider moving citation of ref. 14 from the end of the first (line 2) to end of the third sentence (line 4).”

Response: corrected

Reviewer #3 (Remarks to the Author):

Comment 1: “My main concern is on the 1.6MD hexameric complex (hexamer of the actually solved structure in Fig 2) they presented in Figure 3. They did not provide any experimental evidences to show the existence of such a hexamer. If they can find a way to assemble and stabilize the proposed hexamer and image it directly, it will validate devoting one entire main figure as they presented in Figure 3 and the related portion of main text. Otherwise, the entire Figure 3 and related descriptions/discussions are basically speculations. Then they should tone down the discussion in the main text and possibly adjust related figures.”

Response: The reviewer suggests that we either provide further evidence for the hexamer OR tone down the discussion and adjust related figures. We believe the evidence for hexamerisation are solid:

- 1- All DotL-family AAA+ ATPase function as hexamer. A few AAA+ ATPase purify as monomers but have been shown to function as hexamers in vivo.
- 2- When we generate a model of the DotL hexamer based on the TrwB hexamer, many of the residues mutated in Sutherland et al. (2012) locate at the DotL-DotL interface, thus validating the relevance of this interface in vivo.

However, because Reviewer 3 provides us with the option of toning down the discussion and the related figures, we have modified the text and figure addressing this issue in the following way:

- 1- in Figure 3 (now Fig. 5), we have removed panel a. There is now only one panel (former panel b) that provides a realistic illustration of the hexamer structure.
- 2- In the abstract, we now write: “Six of these hetero-pentameric complex may assemble into a”.
- 3- On page 8, we use the word “indicating” instead of “showing”.

Thus by changing and adding words like “may”, “likely”, or “indicating”, we feel we do not seek to make our case stronger that it may appear. We hope this will satisfy reviewer 3.

Comment 2: “A second concern is Extended data figure 1e. There is a dip in the FSC curves between 7.9 and 5.2 angstrom. This is potentially a warning sign that something is not quite right in either the data processing or collection. The authors should investigate the cause of this.”

Response: Please see response to comment 9 from Reviewer 1.

General Reformatting of the manuscript

The manuscript needed considerable reformatting to comply with Nature communications format. Some extended data figures are now in the main text since Nature communications allows more main text figures. We have added headings throughout. We have incorporated the Methods into the main text. We have provided a single supplementary information file containing supplementary tables, supplementary figures and figure legends, and supplementary references. We also provide an Excell file containing all the primary data as well as the gels.

We would like to thank the reviewers for very useful comments and suggestions that have resulted in a vastly improved manuscript. We hope our manuscript is now acceptable for publication in Nature communications.

Reviewers' comments:

Reviewer #1 (Remarks to the Author):

The authors have provided clear and cogent responses to all the comments in the critiques and made satisfactory changes to the manuscript. One minor typo spotted: p 6, line 6 involve->involved

Reviewer #2 (Remarks to the Author):

Hello, I sincerely apologise for the delay in my reply. I have checked the new version of the manuscript and the authors' reply to the reviewers and I find that all of the points I raised were satisfactorily addressed except for that regarding the functionalities of Cya-DotY and Cya-DotZ constructs, which I repaste below :

Comment 3: "c) Fig 1c: did you check that the Cya-DotY and Cya-DotZ complement their respective single KO mutants? As they are essential to T4SS function, the fact that they are not translocated might otherwise only mean that they are simply non-functional or non-exportable (same logic as point d)."

Response: When testing for effector protein translocation using Cya fusion approach, assessment of the activity of the Cya-effector fusion construct is not required. The reason for this is that effectors are engaged by the secretion apparatus and translocated in an unfolded state, and this is typically mediated by recognition of a C-terminal secretion signal encoded by the effector, which is why the standard protocol is to fuse the Cya enzyme to the amino terminal position in the putative effector. Thus, as long as the fusion construct contains the signal sequence, the Cya enzyme should be delivered into the host cytosol. Because the Cya-DotY and Cya-DotZ proteins contain the entire coding sequence of the Dot protein, the C-terminal signal required for secretion is present.

Let me attempt to make myself clearer here as I think the authors may have misunderstood the point I was trying to make : in figure 1b, the authors suggest that dotZ and dotY are required for secretion of several known T4SS substrates. I say suggest because only secretion in the dotY, dotZ and dotYZ deletion mutants is shown, not secretion in their complemented strains (ie DeltadotZ/pdotZ etc), when the latter is required to show that the secretion defects observed in deltadotZ are solely due to the absence of dotZ etc and not some other genetic defect acquired elsewhere in the deltadotZ strains genome. In this assay, ie in the deltadotZ, dotY or dotZY backgrounds, the cyadotZ and cyadotY constructs tested in 1c for their own secretion only in WT and a deltaT4SS strain (and found negative), really must be tested also because only this control will show whether the cyadotZ and cyadotdotY are functionally WT in terms of allowing T4SS-mediated secretion of other substrates. If cyadotZ does not complement deltadotZ for secretion of other substrates for instance then it is not likely to be fully functional and so that (instead of it being solely a machinery component rather than perhaps a machinery and secreted component all at once as is known to occur for components in several different kinds of bacterial secretion systems) may instead be the trivial reason why its secretion is not observed in the WT background.

I hope that makes my remaining point to the authors, who should otherwise be congratulated for their much improved revised manuscript, clearer.

Please find enclosed the details of our response to the reviewers' comments. Reviewer 1 was pleased with our revisions and recommended publication. Reviewer 2 was pleased with the structural part, but had remaining points to be clarified with the biological part.

Here is how we have amended this revised version in response to Reviewers comments and suggestions.

Reviewer #1

Comment from reviewer

The authors have provided clear and cogent responses to all the comments in the critiques and made satisfactory changes to the manuscript. One minor typo spotted: p 6, line 6 involve->involved

Response

It has been corrected.

Reviewer #2

Comments from reviewer

Let me attempt to make myself clearer here as I think the authors may have misunderstood the point I was trying to make : in figure 1b, the authors suggest that dotZ and dotY are required for secretion of several known T4SS substrates. I say suggest because only secretion in the dotY, dotZ and dotYZ deletion mutants is shown, not secretion in their complemented strains (ie DeltadotZ/pdotZ etc), when the latter is required to show that the secretion defects observed in deltadotZ are solely due to the absence of dotZ etc and not some other genetic defect acquired elsewhere in the deltadotZ strains genome. In this assay, ie in the deltadotZ, dotY or dotZY backgrounds, the cyadotZ and cyadotY constructs tested in 1c for their own secretion only in WT and a deltaT4SS strain (and found negative), really must be tested also because only this control will show whether the cyadotZ and cyadotdotY are functionally WT in terms of allowing T4SS-mediated secretion of other substrates. If cyadotZ does not complement deltadotZ for secretion of other substrates for instance then it is not likely to be fully functional and so that (instead of it being solely a machinery component rather than perhaps a machinery and secreted component all at once as is known to occur for components in several different kinds of bacterial secretion systems) may instead be the trivial reason why its secretion is not observed in the WT background.

Response

The first point made by the reviewer is valid. However, the confusion arises from the fact that we failed to report the important result that as far as growth in amoeba is concerned, **wild type DotY or DotZ completely rescue the deletion of the DotY or DotZ gene**. This is an experiment that we have done and we can confirm indeed that complementation fully rescues the deletions. These results are now presented in Supplementary Figure 2. We have modified the text in the following way (in bold). On page 4, we have added: "**Complementation of the $\Delta dotY$ or $\Delta dotZ$ strains with wild-type *dotY* or *dotZ* gene, respectively, restored growth to wild-type levels (Supplementary Figure 2), indicating that reduction in intracellular growth in the mutants is due to deletion of the targeted gene(s).**" The methods section has also been amended. On page 12, we have added: "**For DotY/ DotZ complementation assays, *dotY* and *dotZ* were cloned into pJB1806 plasmid using InFusion. Wild-type *dotY* and *dotZ* were cloned into the pJB1806 backbone with 200 bp upstream and downstream, so that their native promotor is used for expression.**"

However, we did not do the experiment that consists in monitoring effector secretion in the deletion mutants rescued by wild type DotY and DotZ. The *dotY* and *dotZ* mutants both had measurable defects in their ability to grow intracellularly and to translocate effector proteins into host cells. There was concern over the possibility that these phenotypes were not linked to the deletion of these genes. We have addressed this concern using complementation analysis. These data show that the intracellular replication defect exhibited by the *dotY* and *dotZ* mutants was restored when the wild type gene was returned *in trans* on a plasmid. These data are similar to results obtained for the mutants that are deficient in the coupling protein chaperone proteins IcmS, IcmW, and LvgA. These mutants display relatively minor defects in effector translocation assays that result in more pronounced defects in intracellular replication. Because the intracellular replication defects are the result of decreased effector translocation, complementation studies typically use intracellular replication assays to confirm that there is not a secondary mutation, which confirms that effector translocation defects have been restored. We have added on pages 4 and 5: **“These data are similar to results obtained for the mutants that are deficient in the coupling protein chaperone proteins IcmS, IcmW, and LvgA^{16,17}. These mutants display relatively minor defects in effector translocation assays, that result in more pronounced defects in intracellular replication. Since the intracellular replication defects are the result of decreased effector translocation, complementation studies typically use intracellular replication assays to confirm the absence of a secondary mutation, which confirms that effector translocation defects have been restored.”** References 16 and 17 have been added. Also, on page 5, we have added: **“The Intracellular growth complementation results confirm the decreased levels of effector translocation are due to the genes deletions and not due to secondary mutations.”**

Concerning the reviewer's second point, which is to test the complementation of *cya-DotY* or *cya-DotZ* in the *deltaDotY* or *Z* strains, we are very puzzled with this request. In our view, it is not at all necessary. The experiment is a standard in the field to test whether a protein is secreted or not. The same way we evaluated the secretion of the 5 effectors we show in Fig. 1b (RalF, LegC8, Lem21, CegC3, Lpg1663), we simply tested in exactly the same way *DotY* and *DotZ*, this time asking whether they are secreted. Importantly, this is done in a wild type genetic background so there is no need for the *Cya-DotY* or *Cya-DotZ* protein to be functional. So our opinion is that the *cya-DotY* or *Z* complementation of *DeltaDotY* and *DeltaDotZ* strains, respectively, is not required at all. In this experiment, we ask “Is *DotY* or *DotZ* secreted”. To do so, we do exactly what we do for RalF, LegC8, Lem21, Ceg3 and Lpg1663 i.e. we generate *Cya* fusions of the proteins and test for secretion. This is the way accepted in the field and used to test the secretion of 100s of *Legionella* effectors over the years. The reviewer did not object to this way of doing the test for these effectors, there is no reason to object to the way we tested secretion of *DotY* and *DotZ*.

We hope we have responded to the reviewer's comment in a satisfactory manner and that the manuscript is now acceptable in Nature communications.